# Investigation of Structural Characteristics and Solubility Mechanism of Edible Bird Nest: A Mucin Glycoprotein

**DOI:** 10.3390/foods12040688

**Published:** 2023-02-05

**Authors:** Yating Lv, Feifei Xu, Fei Liu, Maoshen Chen

**Affiliations:** 1State Key Laboratory of Food Science and Technology, Jiangnan University, Wuxi 214122, China; 2Science Center for Future Foods, Jiangnan University, Wuxi 214122, China; 3School of Food Science and Technology, Jiangnan University, Wuxi 214122, China; 4International Joint Laboratory for Food Safety, Jiangnan University, Wuxi 214122, China

**Keywords:** edible bird nest, hydrophobic interactions, crystallization area, solubility properties, water-holding capacity

## Abstract

In this study, the possible solubility properties and water-holding capacity mechanism of edible bird nest (EBN) were investigated through a structural analysis of soluble and insoluble fractions. The protein solubility and the water-holding swelling multiple increased from 2.55% to 31.52% and 3.83 to 14.00, respectively, with the heat temperature increase from 40 °C to 100 °C. It was observed that the solubility of high-Mw protein increased through heat treatment; meanwhile, part of the low-Mw fragments was estimated to aggregate to high-Mw protein with the hydrophobic interactions and disulfide bonds. The increased crystallinity of the insoluble fraction from 39.50% to 47.81% also contributed to the higher solubility and stronger water-holding capacity. Furthermore, the hydrophobic interactions, hydrogen bonds, and disulfide bonds in EBN were analyzed and the results showed that hydrogen bonds with burial polar group made a favorable contribution to the protein solubility. Therefore, the crystallization area degradation under high temperature with hydrogen bonds and disulfide bonds may be the main reasons underlying the solubility properties and water-holding capacity of EBN.

## 1. Introduction

Edible bird’s nest (EBN), the swiftlet’s nest, is a kind of natural food product produced from the saliva of swiftlets of the *Aerodramus* species. EBN has been regarded as a high-grade health food and the choice of functional food as a traditional Chinese medicine for its nutritional and therapeutic values in ancient China. The health benefits [1] related to EBN consumption, such as anti-cough, skin-whitening, nourishing lungs and resolving phlegm, blood circulation, and improving brains and bodies, have attracted considerable attention. EBN extract has been used as one of the popular ingredients in foods, drinks, and nutraceutical products. Currently, EBN is basically purchased as a health food supplement for its high protein and sialic acid content in Asia, especially in China and Malaysia [2], and traded worldwide after harvesting or in processed form.

Proteins, especially glycoproteins with hydrophilic and polar characteristics [2], were found to be the major component of EBN, contributing up to 60% by weight [3]. On the one hand, EBN proteins show a strong water-holding capacity, with swelling multiples of 5.28 to 8.66 after soaking in 60 °C water for 14 h [4]. The strong water-holding capacity maintains the more favorable texture that ensures the quality and consistency of EBN products for customers to taste, such as instant EBN congee, EBN beverages, etc. On the other hand, the poor solubility and low extractive rate of EBN have been a concerning obstacle for further structural research and soluble mechanism exploration in the extraction process. It was reported that the extraction yield of the EBN protein increases under temperatures between 60 °C and 80 °C. However, the extraction yields were generally below 20% under different heating methods, and higher temperatures produced an incomplete break-down of complex proteins [5].

In order to increase the solubility and to further investigate the structural characteristics of EBN, different extraction methods have been applied to extract most of the EBN soluble fraction, while other components remain in the insoluble fraction [5]. According to the traditional consumption method, EBN requires softening by soaking in water for hours and was then heated with distilled water to 60–100 °C and the solubility of the protein was about 37–60% [6]. Moreover, an over-stewing method was modified to enhance the solubility of EBN in water, but the extracted protein was mainly denatured and remained as large proteins [7]. Several other extractants have been employed for the extraction, including salt, alkaline, acid, and enzymatic hydrolysis, but those are unsuitable to acquire the valuable essence of EBN and the active component extractions [8]. Recent studies also investigated that dynamic high pressure micro-fluidization (DHPM) treatment significantly improved the solubility and had effects on the structural properties of proteins in the EBN water-insoluble fraction [9]. However, due to its poor solubility and low extractive rate, further structural research of EBN has reached a bottleneck [10] and protein sequences deposited in the database still remain limited and lack technological advancements. Furthermore, information on EBN’s solubility and water-holding capacity mechanism is not currently available in the literature.

Consequently, an exploration of EBN’s solubility and water-holding capacity mechanism is of great demand. Based on the major components of EBN, we analyzed the solubility properties and structural changes in correlation with the heat temperature from soluble and insoluble fractions. Further discussions were combined with the comparison of the crystallinity to clarify the changeable trend regarding the amorphous or semi-crystalline region and crystalline region of the protein. Thus, this article puts forward an assumption model of the soluble mechanism and possibly explains the reasons for the water absorption and maintenance characteristics of EBN.

## 2. Materials and Methods

### 2.1. Materials and Chemicals

Dry edible birds’ nest samples (*Aerodramus fuciphagus*) originated from Indonesia and were kindly donated by Xiamen Yanzhiwu Sinong Food Co., Ltd. (Xiamen, China). Edible birds’ nest samples with a higher content of protein (66.18 ± 0.76%) were milled by a high-speed universal crusher (SS-1022, Shengshun, Jinhua, China) and then passed through a 120-mesh sieve. A sodium dodecyl sulfate-polyacrylamide gel electrophoresis (SDS-PAGE) kit and protein marker were obtained from Bio-Rad (Hercules, CA, USA). 1-anilinonaphthalene-8-sulphonate (ANS) was obtained from Sigma-Aldrich Co., Ltd. (St. Louis, MO, USA).

### 2.2. Preparation of Extracts of EBN Samples

#### 2.2.1. EBN Extraction

Briefly, 1.0 g of sample was weighed accurately and mixed with distilled water (50 mL) in the tube under the temperature of 40 °C, 55 °C, 70 °C, 85 °C, and 100 °C for 30 min, respectively. Then, the samples were centrifuged at 16,000× *g* for 20 min, and the supernatants and sediments were lyophilized for further experiments.

#### 2.2.2. Protein and Sialic Acid Extraction Rate

Bradford protein assay was used to quantify the protein extraction rate of EBN samples, and the absorbance was measured at 595 nm [11]. The sialic acid content was evaluated based on using high-performance liquid chromatography (HPLC) (Waters 2695) with ZORBAX SB-C18 column (4.6 mm × 150 mm, 5µm). The acetonitrile and water solution (5:95) were used as a mobile phase, with a flow rate of 1.0 mL/min. A fluorescent detector was used, with the excitation wavelength at 230 nm and emission wavelength at 425 nm [5].

#### 2.2.3. Water-Holding Capacity

The free water in the test tube was removed and the weight and volume of the bird’s nest after water absorption and swelling were recorded. The water-holding weighing multiple and water-holding swelling multiple were calculated, and the average of three replicates was taken.
Water-holding weighing multiple= Water swelling weight/sample dry weight(1)
Water-holding swelling multiple= Water swelling volume/sample dry weight(2)

### 2.3. Soluble Compositions Analysis

#### 2.3.1. SDS-PAGE Analysis

SDS-PAGE was carried out according to the methods described by Siti Najihah Zukefli [12] with slight modification. Sixty microliters of protein extract (8 mg/mL) was mixed with 20 μL of SDS sample buffer with 1% β-mercaptoethanol and then heated in a 100 °C water bath for 8 min. Then, the samples were separated using 5% separating gel and 10% stacking gel with a constant voltage of 200 V for 30 min for lectrophoretic analysis. The protein marker with a broad range molecular weight from 10 to 250 kDa was used to estimate the molecular weight of soluble protein distribution under different temperatures.

#### 2.3.2. Intrinsic Fluorescence Spectroscopy Analysis

Intrinsic fluorescence spectroscopy was performed according to the method of Liu with slight modifications [13]. All samples were diluted to the protein concentrations of 0.1 mg/mL using phosphate buffer (pH 7.0, 10 mmol/L). The samples were excited at 280 nm, and the emission spectra were recorded at 300–500 nm with a constant slit width of 5 nm for both excitation and emission.

#### 2.3.3. Surface Hydrophobicity Analysis

Surface hydrophobicity (S_0_) was determined by the method of Sha Huang [14] by assessing fluorescence intensity with fluorescence probe 8-anilino-1-naphthalene sulfonate (ANS). A series protein concentration of 4-mL EBN extractions (0.5, 1.0, 1.5, 2.0, 2.5, 3.0, 3.5, 4.0 mg/mL) were dissolved in 20 mmol/L phosphate buffer (pH 7.4) and quickly mixed with 20 μL of ANS solution (8.0 mmol/L in 20 mmol/L phosphate buffer, pH 7.4), respectively. Then, the fluorescence intensity was measured after 2 min by F-7000 spectro-fluorometer at 365 nm (excitation wavelengths) and 520 nm (emission wavelengths). The constant excitation and emission slit were kept 5 nm. The slope of the fluorescence intensity against protein concentration (mg/mL) plot (calculated by linear regression analysis) was regarded as the number surface hydrophobicity.

#### 2.3.4. Secondary Structure Analysis

The circular dichroism spectroscopy (CD spectra) values of the samples were recorded using a Chirascan V100 machine (London, UK). The EBN extract (300 μL) with a concentration of 0.2 mg/mL was placed inside and the wavelength was set to 190–250 nm [15]. The relative content of secondary structure (α-helix, β-sheet, β-turn, and random coil) was calculated using the CD Pro software.

### 2.4. Insoluble Compositions Analysis

#### 2.4.1. Amino Acid Composition Analysis

The amino acids composition of insoluble EBN were analyzed by high-performance liquid chromatography (HPLC). The liquid chromatographic conditions were referred to Li et al. [15]. Each sample was weighed into a high-temperature hydrolysis tube and added 8 mL 6 mol/L HCl; then, N_2_ was used to seal the tube and it was hydrolyzed at 120 °C for 22 h. After that, NaOH solution was added to the sample to neutralize the hydrochloric acid until the volume was 25 mL, filtered and precipitated by filter paper, and centrifuged at 10,000 rpm for 10 min. We took out 400 μL of the supernatant from each mixture to determine the free amino acids.

#### 2.4.2. Secondary Structure Analysis

The Fourier transform infrared (FTIR) spectrums were collected by Nicolet iS50 FT-IR equipped with an attenuated total reflectance (ATR) adapter, using spectral resolution 4 cm^−1^ and 32 scans [16]. Approximately 20–30 mg of freeze-dried powder samples was placed on the ATR sample compartment and compressed until the reflection was obtained [17]. Spectra were gathered in reflectance mode between 4000 and 400 cm^−1^. Three pellets’ measurements for each sample were performed and FTIR spectra were displayed as reflection values. Band positions related to the secondary structure were calculated using the OMNIC software to support the initial identification of band positions by deconvolution.

#### 2.4.3. Relative Crystallinity Analysis

X-ray diffraction (XRD) of the samples was conducted by an X-ray diffractometer (Bruker, D8 PHASER, Berlin, Germany), with the powder X-Ray diffraction using a copper tube operating at 40 kV and 200 mA and producing Cu-Kα radiation of 0.154 nm wavelength. The diffraction data were collected from 2θ values (5° to 55°) [18], where θ is the angle of incidence of the X-ray beam of sample at a rate of 4°/min and a step size of 0.03° at room temperature. The relative crystallinity (RC) was calculated according to the following equation [19]:RC = (Ac)/ (Ac + Aa) × 100%(3)
where Ac refers to the crystalline peak area and Aa refers to the amorphous peak area. All measurements were performed in triplicate.

### 2.5. Different Combination of Cross-Linking Agents

In order to explain the dissolving properties including strong water-holding capacity and gain a better understanding of the mechanism of the dissolving progress based on bond analysis, different combinations of cross-linking agents were added to the EBN samples, supposing that single solutions of SDS, urea, and β-mercaptoethanol only interrupted the intermolecular hydrophobic interactions, hydrogen bonds, and disulfide bonds, respectively. After treatment under the temperatures of 55 °C and 85 °C, the samples were prepared according to 2.2 to analyze the solubility.

### 2.6. Statistical Analysis

All analyses were performed in triplicate (*n* = 3) and the results were presented as the mean ± SD. The significant difference in means of analyses was determined by analysis of variance (ANOVA) and Duncan test with SPSS (V17.0, SPSS Inc., Chicago, IL, USA). Additionally, the differences were considered significant when *p* < 0.05 was obtained

## 3. Results

### 3.1. Solubility and Water-Holding Capacity of EBN

The heating process played an important role in the solubility of the protein. At higher temperatures, the hydrophobic amino acids are exposed to the environment, making the electrostatic repulsion within the protein molecule increase, and the formation of insoluble high-molecular-weight co-aggregates decreased, thereby resulting in an increased solubility of the protein [20]. The solubility of protein and sialic acid as affected by temperature increase from 40 °C to 100 °C is shown in Figure 1. As shown in Figure 1, the temperature has a remarkable impact on the solubility. As the heat temperature increased, the protein and sialic acid solubility increased from 2.55% to 31.52% and 0.05% to 26.86%, respectively, with an obvious turning point at the temperature of 70 °C. Under low temperature, the EBN protein molecule structure remains stable and represents low solubility. However, under higher temperature, especially over 70 °C, the break of chemical bonds and formation of soluble protein aggregates occurs and probably explains the increase in water solubility [21], which was further confirmed by the results of SDS-PAGE in the follow study.

As a glycoprotein, EBN has a high water-holding capacity. The effect of heat treatment on the water-holding weighing multiple and swelling multiple were investigated and the results are shown in Figure 1. It was observed that the water-holding weighing multiple and swelling multiple of EBN were affected by temperature as well. The water-holding capacity of EBN also increased during heat treatment, which consequently resulted in the increase of weighing and swelling multiple. The results were consistent with the trend of solubility. As the heat temperature increased from 40 °C to 100 °C, the water-holding weighing multiple and swelling multiple increased from 3.83 to 13.67 and 3.83 to 14.00, respectively. It is a common practice to use heat treatments to modify the functional properties of proteins [22]. Proper thermal denaturation of EBN proteins possibly affects interaction with water as well as promotes the solubility without further protein aggregation or coagulation [23].

### 3.2. Structure Analysis of EBN Soluble Fraction

#### 3.2.1. SDS-PAGE Analysis

The composition of soluble EBN protein under different heating temperatures was analyzed using SDS-PAGE under reducing conditions (+β-mercaptoethanol) and the results are shown in Figure 2. The molecular weight distribution of soluble fractions was observed with a wide molecular weight (Mw) ranging from 10 to 250 kDa. It generally consisted of two subunits from 37 kDa to 50 kDa that can be cleaved into low-Mw fragments and two subunits from 100 kDa to 150 kDa that can be cleaved into high-Mw fragments. As shown in Figure 2, the protein banding patterns among the different temperature-heated samples were similar; the reported protein band of 43 kDa, 50 kDa, 108 kDa, and 128 kDa were all observed. However, the intensities of the protein bands of high molecular weight (108 kDa and 128 kDa) gradually increased with the temperature increase process. This result may be attributed to the aggregation of the soluble low molecular weight (43 kDa and 50 kDa) proteins and/or the increasing solubility of high molecular weight proteins at high temperature. A similar phenomenon was also observed [24] and suggested a certain degree of intermolecular disulfide bond cross-linking might result from the glycation process [25].

Further investigation was carried out through different cross-linking agents to verify the contribution of hydrophobic interactions, hydrogen bonds, and disulfide bonds to protein stability. The electrophoresis of the extracted EBN proteins under the temperatures of 55 °C and 85 °C are shown in Figure 3. It was observed in both a and b that in lane 3, re-heated at 100 °C for 30 min, there existed large molecular weight protein-protein compounds that were unable to penetrate the pores of the separating gel. However, samples with SDS and β-mercaptoethanol solution presented a certain depolymerization of larger molecular weight protein-protein compounds into smaller molecular weight protein subunits. This indicated that the protein structure of EBN was mainly supported by the hydrophobic interactions and disulfide bonds, following by hydrogen bonds [26].

#### 3.2.2. Intrinsic Fluorescence and Surface Hydrophobicity (S_0_)

The number of tryptophan residues and their microenvironment can be estimated through the intrinsic fluorescence value [27]. The effect of heat treatment on the conformational transformation and tertiary structure of soluble EBN were further characterized by intrinsic fluorescence and the results are shown in Figure 4A. As the temperature increased, the peak values (λ max) of the samples were decreased with a blue shift about 15 nm at the maximum emission fluorescence wavelength. The significant blue shift indicated the significant structural modifications owing to the heat treatment, which can unfold the protein, and the introduced glucose residues that may affect the polarity of protein. Consequently, the micro-environmental non-polar chromophore of the aromatic amino acid inside the protein was enhanced and further confirmed the significant structural modifications during heat treatment [28]. A similar conclusion was reported by Qunyan Fan [9] to explain the blue shift resulting from conformational rearrangement of EBN proteins.

S_0_ is known as a vital factor that influences solubility and is usually used to evaluate the lever of hydrophobic amino acid residues on protein surfaces and characterize the conformation changes in proteins. Hydrophobic core and distribute hydrophilic tend to be buried in proteins’ native structure and charged amino acids on the surface [27]. The H_0_ of the soluble EBN proteins extracted at different temperatures were measured and the results are shown in Figure 4b. As shown in Figure 4b, the S_0_ values increased from 74.1 to 849.0 when the heat temperature increased from 40 °C to 100 °C, with an obvious turning point at 70 °C. These results are consistent with the change trend of solubility and indicated that more and more hydrophobic groups of EBN were exposed by the heating treatment. It may correspond to the surface exposure of hydrophobic domains originally inside the proteins, which causes protein unfolding and aggregation, thereby increasing surface hydrophobicity [29] and presenting more structural changes of the EBN protein and is possibly related to the improved solubility [30].

#### 3.2.3. CD Spectroscopy Analysis

The secondary structures of soluble EBN proteins in solution were investigated using CD spectroscopy and the results are shown in Figure 5. As shown in Figure 5, the CD spectra of soluble EBN protein samples displayed negative peaks at approximately 212 nm. The increase of heating temperature resulted in a slight increase of the observed negative molar ellipticity, which was considered as an indicator of the loss of the corresponding secondary structural elements [31]. Further results of the contents of secondary structural elements were calculated using CD Pro software. Detailed data of α-helix, β-sheet, β-turns, and random coil contents are listed in Table 1. It was noticed that among the secondary structure contents, the α-helix increased from 8.50% to 33.80%, while the β-sheet decreased from 31.90% to 15.90% after the heat treatment of EBN samples. An explanation could be that the heating process possibly weakens the intermolecular interaction (i.e., the hydrogen bands between hydrogen atoms of amide and oxygen atoms of carbonyl) [32] and results in a reduction of β-sheet. These results were consistent with the FTIR measurements and suggested that EBN extraction preserves the native structure with higher β-sheet contents.

### 3.3. Structure and Crystallinity Analysis of EBN Insoluble Fraction

#### 3.3.1. Amino Acids Analysis

It has been reported that the high degree of glycoproteins and the amount of hydrophilic amino acids may contribute to the solubility improvement during the heat treatment. Consequently, the insights of the basic amino acid composition of the EBN insoluble fraction were discussed to further explore the relationship with solubility [33].

The basic composition of EBN raw material amino acids are listed in Table 2. As a good source of natural proteins, EBN contains all essential amino acids, among which the higher content amino acids in EBN were aspartic acid and valine, with values of 5.31 g/100 g and 4.21 g/100 g, respectively. The most concentrated sulfur amino acid was cysteine, with a value of 0.67 g/100 g, which might contribute to the formation of disulfide bonds during heat treatment. A similar result was indicated by AHLAM ABDASLAM M ALI [34] who also analyzed the major amino acids of EBN.

The components of proteins are important to explain the solubility behavior in solutions because the hydrophobicity of amino acid side chains determines the protein’s solubility [35]. The total amount of the EBN insoluble fraction’s amino acid compositions under different extraction temperature was overall 60%. On the whole, there was no significant change in the major distributions of amino acids. The amounts of acidic amino acids (Asp, Glu), alkaline amino acids (His, Arg, Lys), and aromatic amino acids (Tyr, Phe) have no remarkable difference.

However, as shown in Table 2, most of the non-polar amino acids (Ala, Val, Met, Phe, Ile, Leu, Pro), which accounted for over 20 g/100 g, existed in the insoluble fractions and the heat treatment accelerated the soluble process of polar amino acids (Ser, Thr, Tyr, Cys, Gly), which increased from 7.76 g/100 g to 12.57 g/100 g. In particular, changeable contents of serine and glycine were observed, from 4.03 g/100 g to 3.68 g/100 g and 2.24 g/100 g to 2.16 g/100 g, which can form rich intramolecular and intermolecular hydrogen bonds with alanine, respectively. The soluble mechanism can promisingly be explained by the interactions of solvents with proteins since the amino acid residues of proteins are exposed to the environment during the soluble process [36].

#### 3.3.2. FTIR Analysis

The secondary structure of insoluble EBN heat-treated samples was characterized by using FTIR analysis. It has been reported that the FTIR peaks at approximately 1655 and 1535 cm^−1^ were attributed to amide I and II bands, which were usually evaluated as the most obvious spectral features of proteins [37]. It is observed in Figure 6a that the intensity of the regions of 1650 cm^−1^ (C = O) and 1540 cm^−1^ (C-N) from amide I and II slightly decreased with the increase of the heat temperature.

The spectra in the wavenumber range of 1700–1600 cm^−1^ were analyzed by OMNIC 8.0 software to obtain the area of the resolved peaks and convert the information into the secondary structure contents. The resolved peaks corresponded to the secondary protein structures: α-helices (1646–1664 cm^−1^), β-sheets (1615–1637 and 1682–1700 cm^−1^), β-turns (1664–1681 cm^−1^), and random coils (1637–1645 cm^−1^) [21]; the results of calculated secondary structure contents are summarized in Figure 6b. The changes in these contents indicate the changes in the secondary structure of the protein [38].

The secondary structure of the insoluble fraction of EBN was mainly composed of β-sheets. With the increase of temperature, the content of β-sheets increased from 45% to 49% and the content of α-helix decreased from 25% to 20%. It is estimated that the hydrogen bond that stabilized the protein structure was broken and the EBN protein unfolded; moreover, the content of ordered structures in the protein molecule decreased, causing the molecular structure to loosen [39]. The heat treatment changed the protein secondary structure by breaking the intermolecular hydrogen bonds and increasing protein solubility.

#### 3.3.3. Relative Crystallinity Analysis

X-ray diffraction is usually used for the quantitative determination of various parameters for the characteristics without destroying the sample, and it was employed to examine the structural phase of EBN insoluble fractions; the results are presented in Figure 7. Since the EBN contains a high amount of protein, the X-ray diffraction pattern showed a broad band at 2θ = 9.2° and 2θ = 20.3°, which indicated the partially preponderantly crystalline structure of the samples. As shown in Figure 7, the peak intensity of insoluble fractions increased from 39.50% to 47.81% when the heat temperature increased from 40 °C to 100 °C, which demonstrated the increase in the crystallinity of EBN during the heating process in comparison with raw material.

On the one hand, the gradual increase of major peaks observed indicated that the increase of temperature could induce the degradation of the semi-crystalline and amorphous structure regions of EBN, thereby leading to a looser and smoother structure of protein, which contributed to the increase in solubility [40]. On the other hand, a high degree of crystallinity provided more opportunities for the insoluble fraction of proteins to bind more water molecules, thus performing the stronger water-holding capacity, which is consistent with the previous study in 3.1. The presence and intensity of these peaks is possibly related to the diffraction of the amino acids of the protein surrounding the other functional groups that contributes to making the material more amorphous or semi-crystalline in nature [41]. A similar assumption was raised to suggest that the increase of the amorphous structure of the system will increase the solubility of crystalline substances [13].

### 3.4. Chemical Bond Analysis

Based on the above investigations of EBN soluble and insoluble fractions, it can be concluded that hydrophobic and hydrogen bonding transformation mainly contribute to the solubility of the protein [42]. Further research was conducted to analyze each specific chemical bond based on the protein solubility, on the assumption that the single solution of SDS, urea, and β-mercaptoethanol only interrupt the intermolecular hydrophobic interactions, hydrogen bonds, and disulfide bonds, respectively. Two groups of samples were analyzed under the heat temperatures of 55 °C and 85 °C to contrast the influence of heat treatment on the interaction of different bonding.

Figure 8 showed the protein solubility of EBN samples extracted from different combined buffer systems under 55 °C and 85 °C. Based on breaking specific intermolecular chemical bonds, the solubility of EBN samples in different extraction solutions were observed, which showed that the macroscopic structure of the protein was supported by various chemical bonds [43]. It is clear to compare the specific chemical bond and mutual interactions responsible for stabilizing the structure of extracted samples to analyze in general according to the data of protein solubility in Figure 8. These indicated that increasing temperature could enhance the interactions between hydrophobic interactions and hydrogen bonds and between disulfide bonds and hydrophobic interactions but have little influence on the effect of disulfide bonds. This is also supported by recent published conclusions that the exposure of buried polar groups makes a favorable contribution to protein stability and solubility, which might even be greater than the contribution of non-polar group burial [44]. Consequently, hydrogen bonding and polar group burial make a favorable contribution to protein stability, which also referred as the basis of the extraction of other water insoluble proteins, such as silkworm silk and spider silk protein, due to the existing hydrogen bonds in highly ordered β-sheet forms [45].

### 3.5. Solubility and Water-Holding Capacity Mechanism Assumption

The possible solubility properties and water-holding capacity mechanism of EBN is shown in Figure 9. It was estimated that the EBN protein can be divided into amorphous region and crystalline region stabilized mainly by hydrogen bonding. The higher hydrogen bonded EBN protein with higher crystalline implies a more condensed structure.

At low temperature condition, the amorphous or semi-crystalline region of the protein absorbed water with slight volume swelling and weaker hydrogen bonds were destroyed easily, contributing to the solubility of low-Mw fragments (43 kDa and 50 kDa). Furthermore, at higher temperature, the crystallization region absorbed water to swell, and the volume increased obviously. The crystallization region was destroyed and the solubility of high-Mw fragments (108 kDa and 128 kDa) increased. Meanwhile, heat treatment contributed to the reformation intermolecular hydrogen bond and the exposure of more hydrophobic groups to form hydrophobic interactions and the re-crosslinking of cysteine residues to form disulfide bonds, thus part of the low-Mw fragments aggregated to high-Mw fragments of 108 kDa and 128 kDa.

## 4. Conclusions

In this study, the solubility properties and water-holding capacities were investigated during heat treatment. The solubility and water-holding multiples increased with heat treatment and the intensities of protein banding patterns at different temperature changed during the increase process. For the soluble fractions, intrinsic fluorescence and surface hydrophobicity results indicated that more hydrophobic groups of EBN were exposed. Additionally, the secondary structure was analyzed through CD to observe a reduction of β-sheet and an increase of α-helix. As for the insoluble fractions, most of the non-polar amino acids (Ala, Val, Met, Phe, Ile, Leu, Pro) still existed and secondary structural results from FTIR analysis revealed the breaking of more intermolecular hydrogen bond with the increase of protein solubility. Furthermore, the increase of crystallinity during heat process offered more chances to bind water molecule and resulted in the stronger water-holding capacity. Further research was conducted to analyze each specific chemical bond based on the protein solubility. Finally, the solubility and water-holding capacity mechanism assumption concluded that hydrophobic interactions and disulfide bonds mainly contribute to the aggregation process.

## Figures and Tables

**Figure 1 foods-12-00688-f001:**
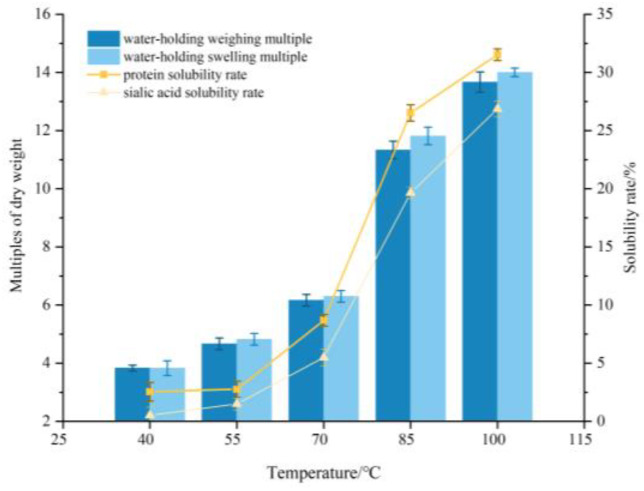
Influence of temperature on water-holding capacity and solubility of edible bird’s nest.

**Figure 2 foods-12-00688-f002:**
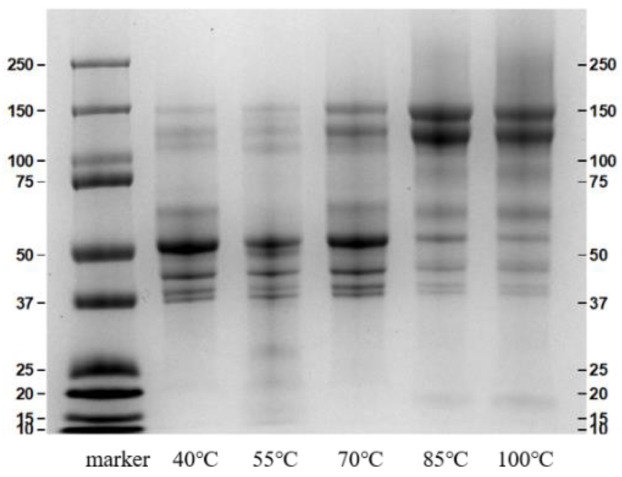
SDS-PAGE analysis of the EBN extracts under different temperature.

**Figure 3 foods-12-00688-f003:**
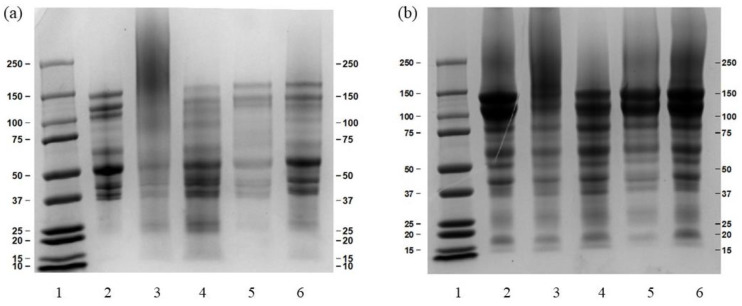
The SDS-PAGE of extracted protein ((**a**) was 55 °C and (**b**) was 85 °C) re-heated at 100 °C for 30 min. 1 were marker; 2 were the extracted proteins; 3 were the extracted proteins re-heated at 100 °C; 4, 5, and 6 were the extracted proteins re-heated at 100 °C with 1% SDS, 8 M urea, and 1% β-mercaptoethanol, respectively.

**Figure 4 foods-12-00688-f004:**
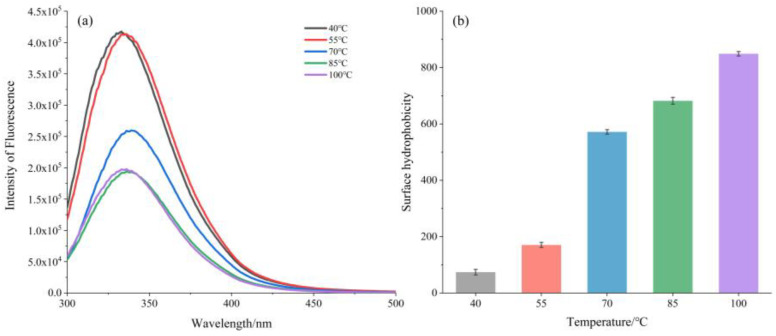
(**a**) Intrinsic fluorescence spectra of EBN treated under different temperature; (**b**) effects of heat treatment on S_0_ of EBN soluble fractions.

**Figure 5 foods-12-00688-f005:**
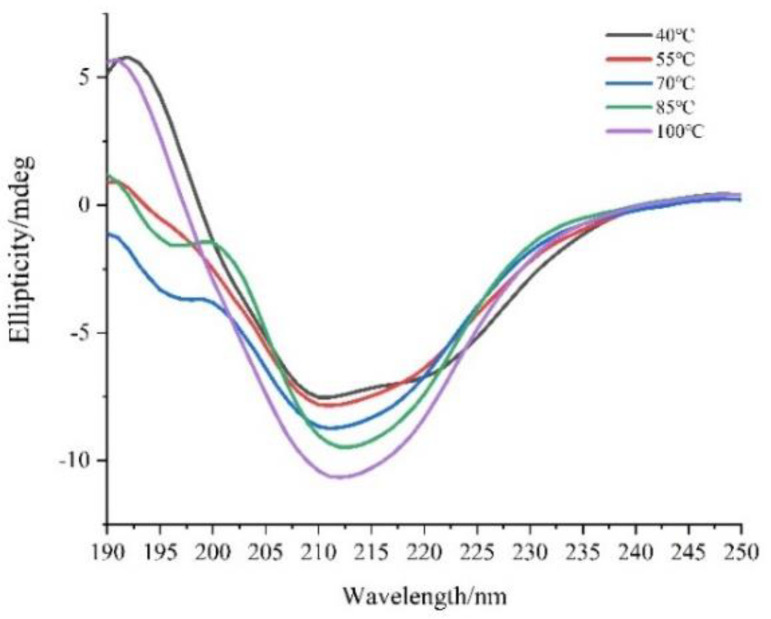
CD spectra of soluble EBN protein.

**Figure 6 foods-12-00688-f006:**
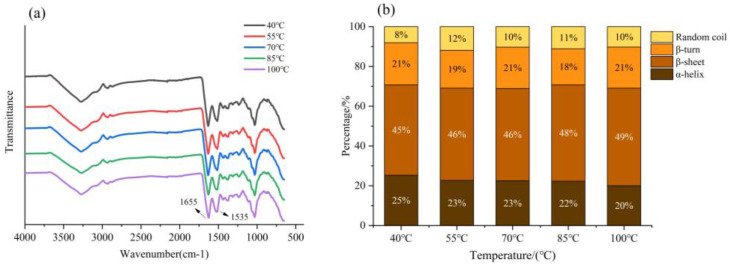
(**a**) FTIR spectrum of EBN insoluble fraction; (**b**) the secondary structure contents.

**Figure 7 foods-12-00688-f007:**
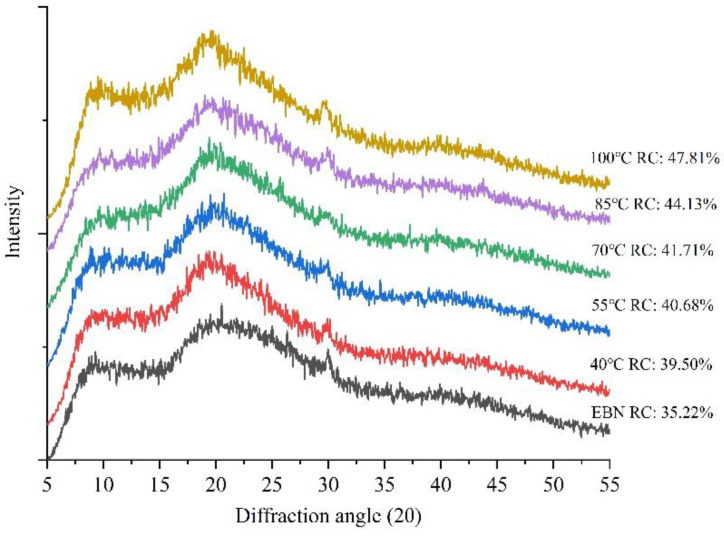
X-ray diffraction of the insoluble fractions.

**Figure 8 foods-12-00688-f008:**
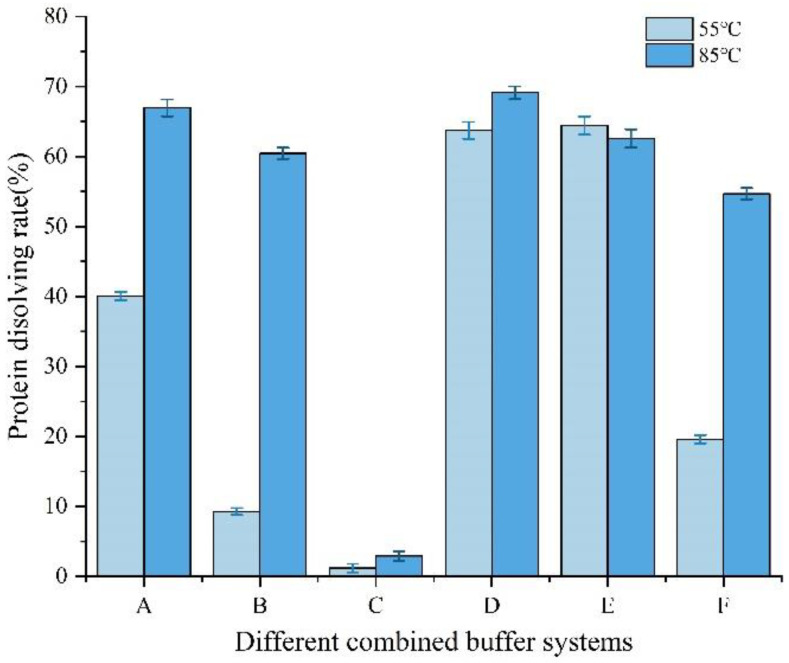
Protein solubility of EBN samples extracted from different combined buffer systems. All of the buffer system contained phosphate buffer solution. A, B and C were added 1% SDS, 8 M urea and 1% β-mercaptoethanol, respectively. D was the combination of 1% SDS and 8M urea. E was the combination of 8 M urea and 1% β-mercaptoethanol. F was the combination of 1% SDS, 8 M urea and 1% β-mercaptoethanol.

**Figure 9 foods-12-00688-f009:**
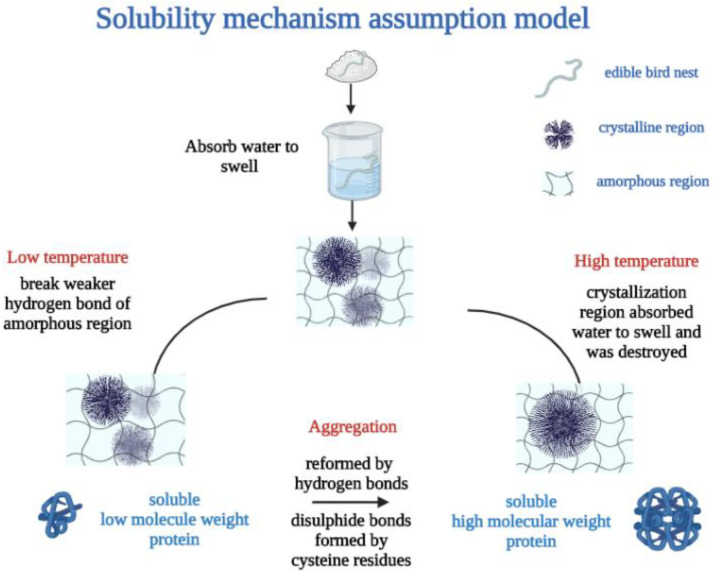
Solubility and water-holding capacity mechanism assumption model of EBN.

**Table 1 foods-12-00688-t001:** Effect of temperature on secondary structure of soluble EBN protein.

Temperature(°C)	Protein Secondary Structure Content (%)				
	β-sheet	Random Coil	α-Helix	β-Turn	NRMSD ^1^
40	31.90	36.30	8.50	23.30	0.053
55	34.00	33.70	8.40	24.00	0.073
70	35.80	31.30	8.80	24.00	0.049
85	27.30	34.10	15.60	23.00	0.094
100	15.90	26.20	33.80	24.20	0.056

^1^ NRMSD (the normalized root mean square).

**Table 2 foods-12-00688-t002:** Amino acid (AA) compositions (g/100 g) of the EBN insoluble fraction.

AA g/100 g	Raw Material	40 °C	55 °C	70 °C	85 °C	100 °C
Asp	5.31	5.72	5.76	5.70	5.60	5.49
Glu	4.26	4.57	4.59	4.56	4.58	4.51
Ser	3.49	4.03	3.78	3.94	3.68	3.75
His	1.98	2.17	2.28	2.16	2.09	2.05
Gly	2.07	2.22	2.24	2.20	2.21	2.16
Thr	3.44	3.82	3.73	3.76	3.67	3.66
Arg	3.60	3.90	3.96	3.89	3.87	3.75
Ala	1.69	1.94	1.78	1.83	1.76	1.78
Tyr	3.04	3.44	3.52	3.43	3.36	3.23
Cys-s	0.67	0.77	0.69	0.76	0.86	0.61
Val	4.21	4.47	4.53	4.47	4.77	4.40
Met	0.55	0.55	0.55	0.03	0.82	0.56
Phe	3.46	3.75	3.79	3.73	3.86	3.60
Ile	1.85	1.97	2.00	1.96	2.07	1.93
Leu	3.84	4.15	4.16	4.13	4.06	4.01
Lys	2.09	2.30	2.29	2.29	2.72	2.23
Pro	3.97	4.28	3.23	4.22	4.03	4.13
Total	56.02	61.23	60.18	60.23	61.23	60.68

## Data Availability

Data are contained within the article.

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
