# Peer review of "Investigation of Structural Characteristics and Solubility Mechanism of Edible Bird Nest: A Mucin Glycoprotein"

_foods, 2023, doi:10.3390/foods12040688_

Round 1
Reviewer 1 Report
Some structural and compositional factors of the bird's nest proteins subjected to different temperatures were studied, with the aim of investigating the mechanisms of solubility and water retention capacity.
In my opinion, the issue of protein structure in relation to its water retention capacity and solubility has been well studied. However, in the case of a protein system such as that of the bird's nest, about which not much information is available.
However, it seems to me that the manuscript has a series of errors and incongruences.
Here I present a list of comments, to provide feedback to the authors:
1) It is necessary to polish the English language, there are a series of grammatical errors throughout the manuscript that does not allow a fluent reading. Even sometimes it is difficult to capture the ideas.
2) In introduction:
L39-43. The phrase "It is commonly..........to taste" is not linked to the one that precedes it.
L41-43. It is contradictory that at the beginning of the paragraph, it is mentioned that EBN consists mainly of glycoproteins with polar and hydrophilic characteristics, in addition to a strong water retention capacity, and that at the end it talks about poor solubility. I think this paragraph should be rewritten, citing references, to justify the idea very well because it is a key point.
L55 and L61-63. Here it is mentioned that there are two fractions: one soluble and one insoluble, something that had not been mentioned before. This must be clarified from the beginning because everything becomes confusing.
In short, the introduction must be substantially improved so that the reader can be convinced of the importance of the research.
3) Methods
L154. How was solubility assessed?
4) Results and discussion
L161-163 and Fig. 1. Here the solubility of sialic acid is mentioned. At what point was sialic acid introduced into the experiment and why? On the other hand, the methodology to assess the water retention capacity was not mentioned. It seems to me that there is a big gap in materials and methods.
L217. The surface hydrophobicity was defined in the methodology as So, but in the results as Ho. It is an error that must be corrected.
L225. Where are these glucose residues introduced and where do they come from?
L283-L285. What does this sentence mean: "The amino acids compositions of EBN insoluble fraction were added up to the total amount over 60%"?
L283. The authors say that overall there were no significant changes in amino acid distribution. Regarding what? Are you referring to heat treatments?
L287. Where is the amino acid composition of the insoluble fraction displayed? Table 2 only shows what corresponds to the soluble fraction.
L287-289. "Heat treatment accelerated the soluble process of polar amino acids". What does that sentence mean? The method to determine the amino acid composition includes total hydrolysis of the proteins, so there would be no reason to expect differences in the amino acid profile between treatments since the same protein extract was used in all of them. The fact that temperature induces conformational changes in a protein does not quantitatively affect the amino acid composition. I think the paragraph is confusing, there needs to be more clarity in the ideas.
L299-301. It is unclear from the spectra that there are differences in intensity in these regions; however, if it exists, an explanation should be given as to why.
L328-331. It is not clear from this discussion whether degraded semi-crystalline and amorphous regions rearrange to later crystallize to give rise to a larger number of crystalline regions. Or is the greater degree of crystallinity at higher temperatures the result of the fact that the crystalline regions are in greater proportion once the amorphous and semi-crystalline regions have been degraded? I really find the whole discussion pertaining to crystallinity analysis confusing.
Author Response
Response to Reviewer 1 Comments
Point 1: It is necessary to polish the English language, there are a series of grammatical errors throughout the manuscript that does not allow a fluent reading. Even sometimes it is difficult to capture the ideas.
Response 1: A series of grammatical errors existed in many compound sentences that used for expression. On the basis of preserving the semantics, the sentence structure was adjusted and the grammatical errors were corrected.
In introduction:
Point 2: L39-43. The phrase "It is commonly..........to taste" is not linked to the one that precedes it.
Response 2: The phrase was changed to “The strong water-holding capacity maintains the more favorable texture that ensures the quality and consistency of EBN products for customers to taste”.
Point 3: L41-43. It is contradictory that at the beginning of the paragraph, it is mentioned that EBN consists mainly of glycoproteins with polar and hydrophilic characteristics, in addition to a strong water retention capacity, and that at the end it talks about poor solubility. I think this paragraph should be rewritten, citing references, to justify the idea very well because it is a key point.
Response 3: This paragraph were rewritten as follow:” Proteins, especially glycoproteins with hydrophilic and polar characteristics[2], were found to be the major component of EBN, contributing up to its 60% by weight [3]. On the one hand, EBN proteins show the strong water-holding capacity with the swelling multiples of 5.28 to 8.66 after soaking in 60 oC water for 14 h [4]. The strong water-holding capacity maintains the more favorable texture that ensures the quality and consistency of EBN products for customers to taste. On the other hand, the poor solubility and low extractive rate of EBN have been a concerning obstacle for further structural research and soluble mechanism exploration in extraction process. It was reported that the extraction yield of the EBN protein increases under temperatures between 60 oC and 80 oC. But the extraction yields were generally below 20% under different heating methods and higher temperature produced an incomplete break-down of complex protein[5] .
Point 4: L55 and L61-63. Here it is mentioned that there are two fractions: one soluble and one insoluble, something that had not been mentioned before. This must be clarified from the beginning because everything becomes confusing.
Response 4: Two fractions: one soluble and one insoluble were mentioned in L47-49.
3) Methods
Point 5: L154. How was solubility assessed?
Response 5: The solubility was assessed by the main components protein extraction rate. The solubility rate was calculated by the concentration of soluble proteion and its total weight in raw material.
4) Results and discussion
Point 6: L161-163 and Fig. 1. Here the solubility of sialic acid is mentioned. At what point was sialic acid introduced into the experiment and why? On the other hand, the methodology to assess the water retention capacity was not mentioned. It seems to me that there is a big gap in materials and methods.
Response 6: Sialic acid, as introduced in L33, was the nourishing substances of EBN and mainly attached to the glycoproteins. The methodology to assess sialic acid and the water retention capacity were added in materials and methods.
Point 7: L217. The surface hydrophobicity was defined in the methodology as So, but in the results as Ho. It is an error that must be corrected.
Response 7: The wrong expression in the results of the surface hydrophobicity were all corrected.
Point 8: L225. Where are these glucose residues introduced and where do they come from?
Response 8: Protein and carbohydrate are the main nutritional components in EBN. Carbohydrate is the second most abundant substance in bird's nest, and as high as 25.6%-31.4%.
Point 9: L283-L285. What does this sentence mean: "The amino acids compositions of EBN insoluble fraction were added up to the total amount over 60%"?
Response 9: It has been changed to “The total amount of the EBN insoluble fraction’s amino acid compositions under different extraction temperature was overall 60%”.
Point 10: L283. The authors say that overall there were no significant changes in amino acid distribution. Regarding what? Are you referring to heat treatments?
Response 10: The conclusion that overall there were no significant changes in amino acid distribution was regarding the heat treatments. It was concluded from the categories and contents of amino acid under different heat temperature.
Point 11: L287. Where is the amino acid composition of the insoluble fraction displayed? Table 2 only shows what corresponds to the soluble fraction.
Response 11: Sorry for the typing error. Table 2 showed the amino acid composition of the insoluble fraction.
Point 12: L287-289. "Heat treatment accelerated the soluble process of polar amino acids". What does that sentence mean? The method to determine the amino acid composition includes total hydrolysis of the proteins, so there would be no reason to expect differences in the amino acid profile between treatments since the same protein extract was used in all of them. The fact that temperature induces conformational changes in a protein does not quantitatively affect the amino acid composition. I think the paragraph is confusing, there needs to be more clarity in the ideas.
Response 12: The polar amino acids (Ser, Thr, Tyr, Cys, Gly) of insoluble fraction protein increased from 7.76 g/100g to 12.57 g/100g while heating. EBN protein consisted of different kinds of submits and some may change during this process. Besides, the amino acid profile was analyzed on the basic of insoluble fractions.
Point 13: L299-301. It is unclear from the spectra that there are differences in intensity in these regions; however, if it exists, an explanation should be given as to why.
Response 13: The peak of 1655 cm-1 (C=O) and 1535 cm-1 (C-N) from amide I and II were marked on the gragh and the intensity of the regions were observed a slightly but not obvious decrease with the increase of the heat temperature.
Point 14: L328-331. It is not clear from this discussion whether degraded semi-crystalline and amorphous regions rearrange to later crystallize to give rise to a larger number of crystalline regions. Or is the greater degree of crystallinity at higher temperatures the result of the fact that the crystalline regions are in greater proportion once the amorphous and semi-crystalline regions have been degraded? I really find the whole discussion pertaining to crystallinity analysis confusing.
Response 14: The results discussed about the relative crystallinity. It was estimated that the greater degree of crystallinity at higher temperatures was the result of the fact that amorphous and semi-crystalline regions degraded during soluble process. And the detailed rearrangement process of degraded semi-crystalline and amorphous regions was further explained based on the chemical bonds.

Reviewer 2 Report
1. English language should be revised
2. Typographical issues have to revise
3. The Editorial issues should check
4. In the introduction, better can present one or 2 sentences as to what are the EBN. What birds are producing?
5. What is the present production of EBN in quantity wise? What is the amount of the economy is produced?
6. What previous studies are reported on the reported topics?
7. Mention the exact research gap
8. Section 2.1, any variety or commercial name is present for the EBN
9. Materials and methods should be a little clear and extended
10. Did you calculate what is the percentage of solubility as the temperature increased?
11. Where is the data analysis? Did the data analyze statistically?
12. Figure 1. Check the legends present in the graph, they are not clear
13. Can you mention depends on the water holding capacity or solubility the EBN can be used in which food applications?
14. What water holding of the EBN will be comparable with what commercial products?
15. Discussion on the relationship between temperature and solubility should be explained in deep.
16. In the presentation of SDS-PAGE need to explain the identified proteins representing the what protein?
17. In 3.2.1, need to work more on the discussion part
18. The Advantages of Surface hydrophobicity of the EBN?
19. Give more discussion on the Surface hydrophobicity of the EBN
20. Section number check for 3.3.3? is it section 3.2.3?
21. There are two headings with “Secondary structure analysis” it is a confusion check once, and need to work more on it.
22. Why do some amino acid concentrations increase and some decrease as the temperature is in raise?
23. Fine-tune the conclusions
Author Response
Response to Reviewer 2 Comments
Point 1: English language should be revised
Response 1: Thank you for your recommendation and the English language was checked again.
Point 2: Typographical issues have to revise
Response 2: Some errors were corrected according to the tamplate.
Point 3: The Editorial issues should check
Response 3: It has been checked and corrected.
Point 4: In the introduction, better can present one or 2 sentences as to what are the EBN. What birds are producing?
Response 4: Edible birds’ nest (EBN), or cubilose, is a kind of natural food product produced from saliva of swiftlets of Aerodramus species. Currently, the main swiftlets producing edible nests are Aerodramus maximus and Aerodramus fuciphagus.
Point 5: What is the present production of EBN in quantity wise? What is the amount of the economy is produced?
Response 5: The world produces over 1,500 tons of bird's nest, with Indonesia producing more than 67% and Malaysia producing more than 23%. Due to its high nutritional and medicinal therapeutic values, EBNs can cost USD 2000–10,000 per kilogram and are regarded as the most expensive animal by-product in the world
Point 6: What previous studies are reported on the reported topics?
Response 6: Previous studies were reported to analyze the structure of EBN protein based on the soluble fraction with low extractive rate. Recently, different extraction methods have been reperted to extract the most of EBN soluble fraction while other components remain in the insoluble fraction.
Point 7: Mention the exact research gap
Response 7: The low extrctive rate of EBN protein exist, of which the solubility about 37-60 % when heated with distilled water to 60-100 oC. Besides, the information on the EBN's solubility mechanism is not currently available in the literature, which may help to increase the extraction rate.
Point 8: Section 2.1, any variety or commercial name is present for the EBN
Response 8: Dry EBN samples (Aerodramus fuciphagus) originated from Indonesia were kindly donated by Xiamen Yanzhiwu Sinong Food Co., Ltd.
Point 9: Materials and methods should be a little clear and extended
Response 9: It has been added to be more clear and extended.
Point 10: Did you calculate what is the percentage of solubility as the temperature increased?
Response 10: Yes, as the heat temperature increased, the protein and sialic acid solubility increased from 2.55% to 31.52% and 0.05% to 26.86%, respectively. That is to say, the protein and sialic acid solubility increased 28.97% and 26.81%, respectively.
Point 11: Where is the data analysis? Did the data analyze statistically?
Response 11: Data analysis has been added in the Section 2.6.
Point 12: Figure 1. Check the legends present in the graph, they are not clear
Response 12: Figure 1 has been replayced.
Point 13: Can you mention depends on the water holding capacity or solubility the EBN can be used in which food applications?
Response 13: It has been added in the introduction L42.
Point 14: What water holding of the EBN will be comparable with what commercial products?
Response 14: In food processing, water holding capacity reflects its ability to prevent water loss. Food with good water holding capacity can maintain excellent taste and character. The holding capacity of comparable commercial products like tremella and agaric is 7.67g/g and 8.67g/g respectively.
Point 15: Discussion on the relationship between temperature and solubility should be explained in deep.
Response 15: More explanation was added in the Section 3.1.
Point 16: In the presentation of SDS-PAGE need to explain the identified proteins representing the what protein?
Response 16: It has been added in the Section 3.2.1
Point 17: In 3.2.1, need to work more on the discussion part
Response 17: More explanation was added in the Section 3.2.1.
Point 18: The Advantages of Surface hydrophobicity of the EBN?
Response 18: The increase in surface hydrophobicity may also correspond to surface exposure of hydrophobic domains originally inside the proteins upon heat treatment.
Point 19: Give more discussion on the Surface hydrophobicity of the EBN
Response 19: More explanation was added in the Section 3.2.2.
Point 20: Section number check for 3.3.3? is it section 3.2.3?
Response 20: It has been corrected.
Point 21: There are two headings with “Secondary structure analysis” it is a confusion check once, and need to work more on it.
Response 21: It has been changed.
Point 22: Why do some amino acid concentrations increase and some decrease as the temperature is in raise?
Response 22: As shown in Fig 2, the molecular weight distribution of soluble fractions were different, thus the composition of insoluble fractions was also different.
Point 23: Fine-tune the conclusions
Response 23: It has been fine-tuned.

Round 2
Reviewer 1 Report
I reviewed the 2nd version of the manuscript and I believe that it complies with the suggested recommendations. Just please cite Table 2 in the text. In my opinion I don't think I have to review it again.
Author Response
Point 1: I reviewed the 2nd version of the manuscript and I believe that it complies with the suggested recommendations. Just please cite Table 2 in the text. In my opinion I don't think I have to review it again.
Response 1: Thank you for your consoderable comments. The Table 2 was cited in L301 and L315.
